# Potential Allergenicity of Plants Used in Allergological Communication: An Untapped Tool for Prevention

**DOI:** 10.3390/plants12061334

**Published:** 2023-03-16

**Authors:** Donát Magyar

**Affiliations:** National Public Health Center, H-1097 Budapest, Hungary; magyar.donat@gmail.com

**Keywords:** allergy, plant, pollen, patient education, visual advertising

## Abstract

Plants are often used to illustrate allergy-related medical products, services, patient information materials and news. The illustration of allergenic plants is an important tool in patient education, contributing to the prevention of pollinosis, as patients can recognize plants and avoid pollen exposure. In this study, it is aimed to evaluate the pictorial content of allergy-related websites depicting plants. A total of 562 different photographs depicting plants were collected using image search, identified and categorized according to their potential allergenicity. Of the total 124 plant taxa, 25% of plants were identified to the genus level and a further 68% were identified to the species level. Plants with low allergenicity were found in 85.4% of the pictures, while plants of high allergenicity were shown in only 4.5% of the pictorial information. *Brassica napus* was the most frequent species identified (8.9% of the overall identified plants), while blooming Prunoidae, *Chrysanthemum* spp. and *Taraxacum officinale* were also common. Considering both allergological and design aspects, some plant species have been proposed for more professional and responsible advertising. The internet has the potential to provide visual support for patient education in allergenic plants, but emphasis must be put on the transmission of the correct visual message.

## 1. Introduction

The pollen of many plant species can cause allergic disease (pollinosis) in susceptible people [1]. In the last two decades, the prevalence of sensitization to pollen increased from 13% to 30% [2]. The most common symptoms of pollen allergy are sneezing, red eyes, runny nose, concentration problems and fatigue. Symptoms can be controlled by pharmaceutical treatments such as local and systemic antihistamines and steroids and by a non-pharmacologic approach in which allergens must be avoided. Patient education often aims at the recognition of allergenic plants to avoid pollen exposure or take protective measures to minimize the health risk. Therefore, allergen avoidance can be considered an essential part of the management of allergic diseases [3].

The allergenicity of plants depends on several factors (such as the plant’s abundance, the pollination strategy and the antigen content of the pollen grains), as a result of which the allergenicity of different plant species can vary widely from non-allergenic to highly allergenic [4]. The information on plant allergenicity should be disseminated and presented in an easily understandable way, preferably with illustrations, as visual information can be more powerful than verbal information [5,6]. In particular, to recognize plants, illustrations are needed [7].

Plants are often depicted in information materials of allergy-related products (e.g., medicines and air purifiers), services (clinics, salt rooms), patient information and news (pollen forecasts). Nowadays, the Web is the most effective medium of advertisement for these contents [8]. According to a study performed in Poland, approximately 76.9% of adult Internet users (N = 1000) go to online platforms to obtain medical information [9]. A questionnaire survey found that most of the people affected by ragweed allergy follow pollen information (93%), mostly by websites and television, but a considerable amount of them (18%) are not interested in the source of information [10]. A study on the YouTube videos on allergic rhinitis showed that a total of 36% of the videos can be classified as misleading, while 43% can be classified as useful [11]. The lack of content regulation (e.g., peer-review) on a vast number of websites make it difficult for the patients to critically distinguish the information. It is well known that picture-based webpages have a better advantage in advertising attention versus text-only content [12]; thus, plant pictures are intensively used on websites dedicated to pollen allergy treatment and prevention. Pictorial information of plant allergenicity has not been studied before in detail, although other types of visual information with an impact on the patients have been frequently analyzed (e.g., medication instruction labels, pictograms and infographics, e.g., [13,14]). Due to the extensive use of pictures, it is timely to evaluate the pictorial information of plant allergenicity provided by websites. The aim of this study is to evaluate whether the images provided by allergy-related websites are suitable for conveying appropriate information about the allergenicity of plants.

## 2. Materials and Methods

Pictorial information on plant allergenicity was collected from the Internet from May 2021 to July 2022 using the image search engine of Google (for research with a similar strategy, see, for instance, [15]). The Boolean operator ‘AND’ was used to combine the following keywords: ‘pollen’ AND ’allergy’. Pictures were manually examined to ensure that multiple photos of the same series of illustrations were removed.

Plants found on the pictures were identified using botanical and horticultural literature. The identified plant taxa were categorized by their potential allergenicity using the CARE-S index [16]. In this index, categories are calculated as a function of genetically determined factors of plants: immunogenicity (Im), morphology (Mo) and pollen production (Po). The allergological literature was reviewed for IM. The morphology of male flowers and pollen is considered to categorize plants according to their pollination strategy. Light, small (usually >40µm) and dry pollen grains with a low specific gravity, or with air sacs with a (nearly) smooth surface, were considered to be anemophilous. The pollen production of cultivars with a high or reduced pollen yield was also considered in the categorization. For a detailed description of the Im, Mo and Po parameters, see [16]. The combination of these parameters yields the following formula:AC_p_ = Im × Mo × Po
where AC_p_ is a raw value of the potential allergenicity which is converted to an index with five categories, referred to as the CARE-S index (CAtegorization System for REgulation of Allergenic Plants by Strong Evidence) (Table 1).

In order to evaluate the role of the plants in the ads, their co-occurrence with other visual components of the pictures (such as depicted patients, allergic symptoms and medicines) were also analyzed. Additionally, the qualitative relationships of plants with other visual elements were assessed with formal analysis, a technique used to evaluate visual elements and principles of design [17,18]. The most common pictures were reverse-image-searched for counting occurrences using the ‘Google Search by Image’ function on Google ([19], https://images.google.com/ (accessed on 8 August 2022)), as this technology offers an opportunity to rapidly collect picture-based information [20]. To see how frequently plant illustrations are used on allergy-related websites, another Google search was performed using the words ‘pollen’ AND ‘allergy’. The websites are grouped according to their main function (daily news, healthcare magazines, clinics, products, professional associations). Only one illustration per website was considered: the first one of the results of the image search.

## 3. Results

A total of 562 different photos depicting identifiable plants were found on allergy information websites. In the pictures, 124 plant taxa, belonging to 41 plant families, were identified; among them, 25% of the plants could be identified to the genus level, and a further 68% could be identified to the species level, representing taxa from various biogeograpical regions. Most plants (32.7% of the cases, 29 species) belonged to the Asteraceae family, followed by Rosaceae (19.2%, 9 spp.) and Brassicaceae (3.7%, 3 spp.). Fabaceae was a less common but diverse family containing 11 species (Table 2).

The most frequently referred taxa were Prunoidae, *Brassica napus*, *Chrysanthemum* spp. and *Taraxacum officinale*, which, together, made up 34.5% of the identified plants. *Brassica napus* was the most frequent species identified, forming 8.9% of the overall identified plants. Mostly, herbaceous (71.0%) taxa were illustrated as allergenic plants, while woody taxa were less frequently depicted (29.0%). Catkin-bearing plants (*Alnus*, *Betula*, *Corylus* and *Salix* spp.) have been used rarely (3.6%). Colorful flowers (96.0%) were preferred, while inconspicuous, green flowers, e.g., of grasses, were relatively uncommon (4.0%). The dominant color on the pictures was yellow (38.9% of the pictures, 21.8% of the taxa), mostly depicting *Brassica napus*, *Taraxcum officinale* and *Forsythia* × *intermedia*, followed by a white color, being dominant in 30.9% of the pictures (represented mostly by Prunoideae). Pink was third on the color preference list, being in 14.5% of the total pictures, often depicting ornamental taxa belonging to Prunoideae. Other flower colors appeared less frequently in allergy-related marketing (purple 7.5%, red 6.3%, green 3.9%, orange 2.2%, blue 1.0%).

Potential allergenicity (CARE-S index) was applied to categorize the identified plants. For herbaceous taxa, the CARE-S index was not available in the literature, because former publications focused on urban trees. Thus, the CARE-S index was calculated for 106 taxa for the first time (see Appendix A). In 85.4% of the allergy-related pictures, plants with low allergenicity (CARE-S 0 and 1) were found. As some picture contained more than one plant species, the frequency of the allergenic plants was also calculated. Only 3.2% of the plants belonged to highly allergenic species (CARE-S 4), vs. 79.8% of the plants belonging to low-allergenicity taxa. Among non-allergenic (CARE-S 0) taxa, the most common plants (13.0% of the total taxa) belonged to the subfamily Prunoideae, containing ornamental trees and shrubs, such as *Prunus serratula* ‘Kanzan’. In the CARE-S 1 group (low potential allergenicity), *Brassica napus*, *Chrysanthemum* spp. and *Taraxacum officinale* were the most frequently presented plants (having 8.9, 6.4 and 6.2% of the total taxa, respectively). Among allergenic plants, Poaceae (mostly *Dactylis glomerata*, CARE-S 2), *Ambrosia artemisiifolia* (CARE-S 4) and *Solidago* spp. (CARE-S 2) were the most frequently illustrated (in 2.3, 2.1 and 1.4% of the total taxa, respectively), while other common allergens (*Alnus*, *Betula*, *Corylus* spp.) were rarely shown (<0.9%). *Forsythia* × *intermedia* was the most frequent species in the illustration materials (2.3% of the total taxa) with unknown allergenicity (i.e., no sufficient information to calculate the CARE-S index). Figure 1 shows the potential allergenicity of plants belonging to different plant families.

Based on the Google search, 63.0% of the allergy-related information on the Internet (N = 108 websites containing the words ‘pollen’ AND ‘allergy’) was illustrated with plants. The images have been found as a prominent visual element in advertisements for various allergy-related products, services of the clinical and pharmaceutical sectors, alternative medicine and products such as air purifiers. Plant images were also used in the news, i.e., as illustrations of pollen information in daily news or in articles on news websites. In detail, 37.8% of the plant illustrations were found on websites of clinics. Daily news and healthcare magazines were the source of 22.5% and 19.8% of plant illustrations, respectively, while products (11.7%) and professional associations (8.1%) contributed to a lesser extent. A total of 82.0% of the webpages presented misleading pictorial information by showing non-allergenic plants with allergic persons. The improper presentation of plants was found in 31 of 42 clinics, in 22 of 25 daily news sources, on 12 of 13 webpages offering products, in 20 of 22 healthcare magazines and in 6 of 9 professional associations.

Stock photography websites served as major suppliers of most plant photographs licensed for the above-mentioned specific uses. The plants and patients with allergy symptoms are often illustrated together (N = 403), and almost all of them (99.0%) demonstrated an allergenic effect of the illustrated plant by blowing nose (43.3%), sneezing (35.8%), running nose or having irritation of the nose (5.5%), red eyes (3.0%) or wheezing (0.3%); they alleviated allergy by wearing a surgical mask (7.3%) or gas mask (3.0%), applying a nasal spray/drop (0.8%), using an inhaler (0.3%) or taking pills (0.5%). Among the depicted patients (N = 353), mostly young women (70.5%) were illustrated, while young men (11.0%), girls (10.5%), boys (4.2%), elderly women (3.1%) and elderly men (0.6%) were shown in a lesser number. In the most popular picture, having 886 search results, a young woman sneezing on a dandelion (*T. officinale*) field was shown. This picture appeared in different languages in allergy-related advertisements (36.0% products and 18.0% clinics) and news (36.0% daily news and 10.0% healthcare magazines, among 50 websites analyzed for content) and was awarded with a high usage score [21].

With the formal analysis of pictures (N = 80), the artistic role of the plants in the ads and their relationship with other visual components of the pictures (such as depicted patients, allergic symptoms and medicines) were assessed. The visual elements and design principles often used in this type of illustration are recognizable (Appendix A). The role of the visual elements was to draw the viewer’s attention to the central issue of the advertisement, i.e., the allergic reaction of the patient. The most pronounced visual elements were shapes and lines, color, value, space and composition. A combination of visual elements can be used to build up the principles of design, which aim to create effects, i.e., feelings in the viewer. According to this, two major design concepts can be identified in the allergy-related ads. The first type of visual advertising concept gives the impression that the patient is surrounded by and is ‘sinking into’ a large field of ‘allergenic’ plants (Figure 2). The second type of visual advertising concept conveys the feeling that the ‘allergenic’ pollen is ‘falling onto’ the patient, usually from trees (Figure 3). The first and the second types of concepts were presented in 58% and 42% of the pictures, respectively.

## 4. Discussion

### 4.1. Plants in Ads: The Botanists’ Perspective

In this study, predominantly (85.4%) non-allergenic plants were found on the Web, mainly as visual advertising materials of allergological products and services. This result indicates that the plants have not been used properly by the designers or those who are responsible for the content, as their allergenic effect has not been taken into account.

*Brassica napus* (rapeseed or canola) was the most frequently depicted species as an allergenic plant in marketing materials, but it has a low potential allergenicity. The illustration of this plant in allergy-related information materials is not only misleading but can have negative effects on other sectors, such as agriculture. Rapeseed represents a large segment of the edible oil and biodiesel market; its production was estimated at 58.4 million tons/year [23]. Undoubtedly, rapeseed is preferred by designers, probably because of its attractive yellow flowers. Rapeseed, like many other non-allergenic plants with colorful flowers, provides a habitat and food for honeybees and other endangered pollinators. *Helianthus annuus* (sunflower), another important crop plant, was also illustrated as an allergenic species, despite its low relevance in pollen allergies. Its pollen has a low concentration in the air, except during harvesting, as demonstrated by a study reporting symptoms caused by sunflower [24]. However, under natural conditions of pollen release, the high potential allergenicity cannot be attributed to the sunflower. Prunoidae were the most common non-allergenic taxa depicted as an allergen. Some species belonging to this plant family (*Pyrus calleryana*, *Malus trilobata*, *Prunus padus*) are optimal candidates for creating a low-allergen garden or urban green spaces, especially *Prunus serrulata* ‘Kanzan’, the protagonist of Japan’s cherry blossom festival hanami. Dandelion (*Taraxacum officinale*, Asteraceae) is an insect-pollinated weed. This plant was often depicted in marketing materials, triggering allergy by its cypselae fruits (‘blowballs’) and wind-blown *pappus* (which aids in seed dispersal). However, these organs do not contain any pollen. The *pappus* is too large (diameter ~7 mm) to be aspirated [25]. Blowballs may attract customers’ attention toward allergy-related presentations, creating inadequate associations. This plant appears in 39 and 38 allergy-related advertisements and news sources, respectively, and in the most popular picture, with 886 search results. *Chrysanthemum* spp. has often been mispresented as an allergenic plant, but it is a popular flower crop with high commercial importance. The potential allergenicity of this plant as well as other decorative herbaceous taxa (such as *Dianthus*, *Gladiolus*, *Gypsophila*, *Lilium*, etc.) is usually low, as their allergic effect is rarely reported in outdoor environments or homes. It is worth mentioning that these taxa can cause occupational allergies, mostly in florist workers [26,27,28,29,30,31,32]. These studies show that florists not only experience respiratory allergies but also dermatitis due to skin contact with handled plants [33]. Besides pollen allergens, triggers of florists’ symptoms include sesquiterpene lactones [34] and parasites of the decorative plants [35]. Neither skin allergy nor parasites are considered in the current study, which focuses on pollen immunogenicity of the respiratory system [16]. It is therefore necessary to maintain the position that the portrayal of the decorative flowers as allergens is misleading. Although information materials on allergenic plants that transfer knowledge to patients are available [36,37,38,39], they were largely omitted from the studied illustrations, possibly because of the different perspective of advertising professionals and designers.

### 4.2. Plants in Ads: The Designers’ Perspective

The primary question for the designer is what makes an image successful in allergy-related advertisements. Apparently, perceived flower beauty is influenced by flower shape and color [40]. A simple Google image search for the keyword ‘flower’ results in colorful, large flowers in 98.8% of images (N = 500, daisies, roses and lotuses, in most cases). This demonstrates our primary association or archetype regarding the word ‘flower’. As a consequence, when one talks about flowers, most people think of colorful, large flowers rather than catkins or other inconspicuous inflorescences of wind-pollinated, allergenic plants. Professional designers may not be an exception. In guidelines for designing symbols, the essential characteristic of the icon ‘flower’ is represented by a daisy [41]. This symbolism may also contribute to the frequent (and possibly often unconscious or ignorant) misuse of large, salient, non-allergenic flowers in the illustration of allergy-related content on the internet.

As this study focuses on plants, their main visual feature, color, is discussed in detail. Color, as a functional component of human vision, can capture attention and contribute to the success of an advertising campaign of a product or a service [42,43]. Increasing the salience of a given concept leads to increased activation of the parts of the brain devoted to processing it [44] due to the role of salience in the process of expectation-shaping [45]. Visual salience is important in medical advertisements, too [46]. The success of the application of yellow-colored plants in allergy-related marketing materials can be explained by the visual priority of yellow relative to luminance matched colors at opposing ends of the wavelength spectrum (i.e., red and blue) [47]. Against these colors, yellow was consistently seen as occurring first in the laboratory tests performed by Hu et al. [47]. According to this study, there is a temporal priority in processing yellow among competing colors, indicating its special salience rooted in the attentional and motivational regulatory functions. In the allergy-related marketing pictures, blue is often pitted against yellow e.g., when the background of a rapeseed field (yellow flowers) is contrasted to the sky (blue). A strong contrast also appears against dark green when yellow flowers (e.g., dandelion) are depicted on a meadow. Studies have shown how yellow, blue, green and red carry various meanings for different but related pharmaceutical products [48,49]. It was found that sore throat medicine is preferred in yellow and green [50]. Apparently, the color of marketing materials for allergy medicines has not been investigated in detail. However, according to the present results, it seems that salient colors of the flowers, especially yellow, are thought to be attractive to the customers of medical products and services in allergology. Possibly, costumers associate yellow with pollen and honey [51], while blue represents air, freshness, cleanliness and, consequently, breathing clean air [52].

Some components with a high frequency in the pictures were identified by the Web-based search. The most frequent components found in these pictures besides plants are: allergic person (99.0%), represented by a young woman (70.9%), blowing nose (43.3%) or sneezing (38.9%) and standing on a field surrounded by yellow flowers (38.9%). Interestingly, all of the above-mentioned properties can be found in the most popular picture, ‘Woman sneezing with tissue in meadow’, having 886 search results [21]. Using the technique of formal analysis, major elements (color, lines and space) and principles of design (contrast, movement and emphasis) could be identified in the pictures that are popular in the marketing of allergy-related products. The best compositions put emphasis on the patient (by positive space and contrast) and plants (by color and contrast) and contain a dynamic expression of patients’ allergic reactions to plants (Figure 2 and Figure 3). By an artistic use of contrast and lines (displayed by the branches of the plants, the patient’s arms, the shape of a handkerchief, etc.), the viewer’s eye is directed to move from the trigger (flowers) to areas of emphasis (respiratory organs, mouth and nose). Allergic symptoms and their strength are exhibited by dynamic movements, among which the most expressive is sneezing. Among symptoms, blowing nose was more frequently illustrated than sneezing, probably because photo models can more easily imitate it than sneezing. In the pictures with high search results, the artistic tools—whether used consciously or intuitively—were able to enkindle impressions, feelings or ‘somatic understandings’ (according to [53]), as was shown by the visual ads applying concepts of ‘falling on’ (tree pollen falling onto the patients) or ‘sinking into’ (patients sinking into a field of allergenic weeds). Apart from the obvious associations of some design elements with particular actions, emotive sensations were produced by certain linear patterns, e.g., lines directing the eye downward evoke moods of sadness or defeat [54].

The application of medicines, pills, nasal drops and inhalers was rarely shown, as this may limit the usability of the picture in advertising a wider variety of medical products and services. On the other hand, plants seem to make an important contribution to the success of pollen allergy-related ads, as they are major visual elements of the advertisements.

### 4.3. Misinforming by Illustrations: An Unnoticed Trap

The misuse of plant pictures is possibly rooted in Plant Awareness Disparity [55], often called ‘plant blindness’ [56], a term used for ‘the inability to see or notice the plants in one’s own environment’. Plant Awareness Disparity leads to, *inter alia*, (i) the inability to recognize the importance of plants in human affairs; or (ii) the misguided, anthropocentric ranking of plants as inferior to animals, leading to the erroneous conclusion that they are unworthy of human consideration [56,57]. Research results indicate that people are more and more ignorant to pants and can no longer identify them [58]. Recognizing that plants have an impact on our health can help to increase awareness of plants, as was demonstrated in a study testing the recognition ability regarding toxic plants [59]. According to the tests, toxic plants were more rapidly detected both by children and adolescents than non-toxic plants. Unlike toxicity, which is mostly related to fruits, allergenicity is related to flowers. The recognition of toxic plants was crucial for the survival of our ancestors for thousands of years, while pollen allergy is a relatively new issue. Thus, the risk perception of flowers is probably not developed in humans, unlike the risk perception of toxic fruits, which is—according to the habitat selection theory [40]—controlled by color preference, a heritage of the past that is hardwired in our brains. Flowers have aesthetic rather than survival-relevant value in human culture [40].

Pictures have an important role in health education, where they can facilitate conceptual processing [60,61]. The present study has shown that the current imagery in allergy-related advertisements has little or no potential for health education for allergy sufferers. Similar to the current study, Remvig et al. [11] concluded that Internet-based information of allergic rhinitis is often misleading. The quality of YouTube videos on allergic rhinitis does not correlate with the popularity of the video. Nonprofessional YouTube channels publish the most popular content; meanwhile, they are responsible for the most misleading videos, whereas associations tend to be the most reliable source of information. Although information about allergenic plants is crucial in primary prevention (i.e., to avoid allergen exposure) [62], more than 50% of allergy sufferers have no or only vague recognition of the plant species causing their symptoms (Szigeti et al., unpublished). Plant allergenicity information is available mostly on websites but also on printed materials. Journals, posters and leaflets placed in healthcare institutions such as pharmacies and allergologists’ consulting rooms are mostly provided by governmental and non-profit organizations (NPOs, NGOs and CSOs) aiming to improve public health. Profit-oriented firms (e.g., pharmaceutical companies) also have an important role in the prevention of allergic diseases. Their multimedia advertising campaigns have enormous potential to raise public awareness and communicate health information. Direct-to-consumer advertising is a particularly powerful tool, having benefits for providing quick and accessible health information to the public [63]. However, it is important that health communicators and policy officials carefully consider advertising’s role in health contexts, monitor the dynamic regulatory environment and develop a deeper understanding of consumers’ perceptions and uses of pharmaceutical advertising [64]. The information distortion and extraneous decoration of pharmaceutical advertisements were often criticized [65,66]. According to Knoesen et al. [67], pharmaceutical advertisements often contain irrelevant and false information that presents potential risks if misunderstood and/or misinterpreted. Othman et al. [68] concluded that the low quality of pharmaceutical advertisements is a global issue. Since these observations were published in 2009, there is still a conflict between promotion and information in the pharmaceutical industry [69]. Although most of these critical studies focus on written information in advertisements, visual information seems inaccurate and misleading as well.

Most consumers cannot perceive the incongruity between pictorial and textual information as mismatching [70]. When verbal and pictorial information are at a mismatch, people tend to rely on the pictorial information [71]. This phenomenon is explained by the well-known picture superiority effect on memory (i.e., the advantage of information presented as a picture over words; see [72]). Moreover, information that is cognitively easier to process is more likely to be accepted as true [73]. Compared to texts, images attract attention more easily and are processed more quickly [6,74], as understanding images requires less cognitive effort than words, i.e., images are processed in a more unintentional and unconscious way [75,76]. The depicted properties of advertised products are more accessible in consumers’ mind [74] and more easily stored by memory [77]. On the other hand, pictorial information is found to be highly influential in the false memory creation process [78]. It seems that misinformation in visual communication is particularly insidious (i.e., it is less observable but all the more damaging) and therefore deserves more attention.

### 4.4. Integrating Art and Botany—A Proposal for a Successful Strategy

As was demonstrated, the depiction of non-allergenic plants in allergy-related information or marketing materials is misleading, but in a correct form, it can be beneficial, providing useful knowledge to the patients. As designers are not necessarily health experts and health professionals are not necessarily trained to produce visual marketing materials, multidisciplinary collaboration is required [79]. By combining design and botanical knowledge, a correct and more targeted marketing strategy can be developed. Colors are obviously effective marketing tools; thus, colorful and allergenic plants should be selected for illustrating allergy-related marketing materials. The list of plants suggested for this integrated marketing strategy is shown in Table 3 and Figure 4.

Despite their attractive, yellow catkins, some allergenic plants, such as *Betula* and *Corylus*, appeared surprisingly infrequently (0.9 and 0.7%) in marketing materials. Illustrations on catkins can preferably be published in spring, considering the pollination period in the temperate region. *Solidago*, another plant with attention-grabbing yellow flowers, can cause symptoms in summer and autumn. In warm, temperate regions, *Acacia dealbata* is a major allergen plant [80]. It is also a noxious invasive plant. Illustrations of its beautiful, yellow flowers can be effectively used for both preventive patient education and marketing purposes in geographical regions where this plant threatens the natural flora through invasion.

Common ragweed (*Ambrosia artemisiifolia*) deserves distinct attention, as it is considered to be the most important allergen in many countries where its eradication is a high-priority task (e.g., Austria, Canada, France, Hungary, Italy, Serbia and the USA). Eradication campaigns and the recognition of this plant are often promoted by advertisements financed by the local governments (e.g., [81,82]. Some information materials also aim to educate the public regarding distinguishing ragweed from ornamental plants with similar leaves, e.g., *Tagetes* spp. and *Chrysanthemum vulgare* [83]. As a success of proper visual information materials, ragweed was recognized vs. the similar plants by 55.6, 68.0 and 75.6% of children, belonging to 6–10, 11–14 and 15–18 age groups [84]. Unfortunately, ragweed is less suitable for visual marketing (having small, inconspicuous inflorescence), but its display on commercial products would transmit a message that the company is supporting public interest objectives. As ragweed is an invasive weed species, raising awareness of this plant has a positive impact on agriculture, too.

## 5. Conclusions

In the marketing of allergological products, there is a dilemma: attract visitors’ attention (i.e., using bright colors of non-allergenic flowers) or convey correct information (i.e., show relevant, i.e., allergenic, but less attractive flowers)? This research has shown that plant imagery in advertisements has little or no potential for health education for allergy sufferers; in fact, it is rather misleading. More professional and responsible advertising can contribute to more informed patients. To take advantage of the opportunity of using plant pictures in allergy-related advertisements, designers need a better understanding of allergenic plants, and health professionals should collaborate with them. To produce visual communication materials that are of the most benefit to patients and take into account marketing aspects as well, it is suggested that advertisers use colorful and allergenic plants for illustrating allergy-related marketing materials. These practical implications are of interest for both allergologists and designers, as they contribute to the development of a more focused health education and marketing strategy. Correct illustrations will help patients recognize and avoid allergenic plants and reduce exposure to allergenic pollen.

## Figures and Tables

**Figure 1 plants-12-01334-f001:**
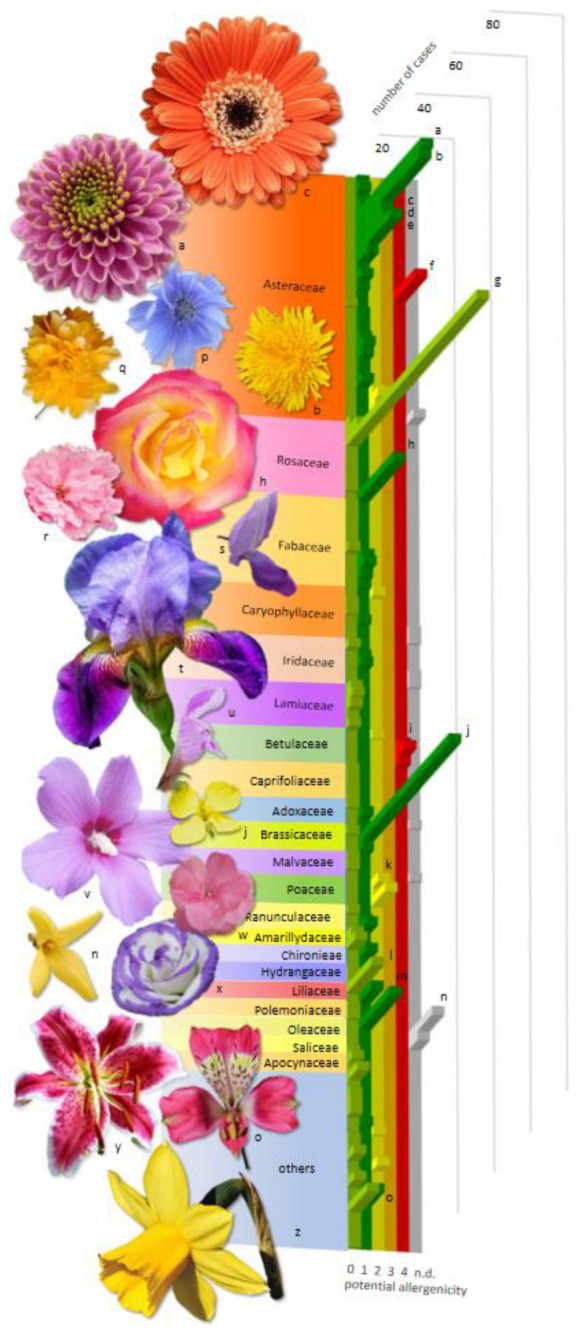
Plant taxa identified in allergy-related marketing materials. On the right side, a graph shows the number of cases (occurrence of plant species in allergy-related marketing materials) in allergenicity groups 0–4 plus n.d. (light green–gray graph bands). The potential allergenicity of plants is expressed by the CARE-S index [16], where 0: non allergenic (light green), 1: low (dark green), 2: medium (yellow), 3: high (orange), 4: very high (red) potential allergenicity, n.d.: no data (gray). The names of individual plant species are not shown (for this, see Table 2), but they are grouped according to plant families. Some remarkable species, however, are highlighted on the graph (a–o). a: *Chrysanthemum* spp., b: *Taraxacum officinale*, c: *Gerbera hybrida*, d: *Leucanthemum vulgare*, e: Asteraceae, f: *Ambrosia artemisiifolia*, g: Prunoideae, h: *Rosa* sp., i: *Betula* spp., j: *Brassica napus*, k: *Dactylis glomerata*, l: *Tulipa* spp., m: *Syringa vulgaris*, n: *Forsythia* × *intermedia*, o: *Alstroemeria ligtu*, p: *Cichorium intybus*, q: *Kerria japonica*, r: *Prunus serrulata* ‘Kanzan’, s: *Wisteria sinensis*, t: *Iris* × *germanica*, u: *Lamium purpureum*, v: *Hibiscus syriacus*, w: *Nerium oleander*, x: *Eustoma grandiflorum*, y: *Lilium* sp. (hybrid), z: *Narcissus pseudonarcissus*. Some examples of plant species are shown in the photographs on the left side (a–z; for species names, see above). Source of the images: copyright-free photos from pixabay.com (a, j, n, o, t, u, x–z), unsplash.com (c) and freepik.com (r). Some photos were taken by the author (b, h, p, q, s, v, w). The photos have been edited to clearly illustrate the flower morphology of the plants with nil or low potential allergenicity (individual flowers separated from the plant, background removed, brightness and contrast adjusted).

**Figure 2 plants-12-01334-f002:**
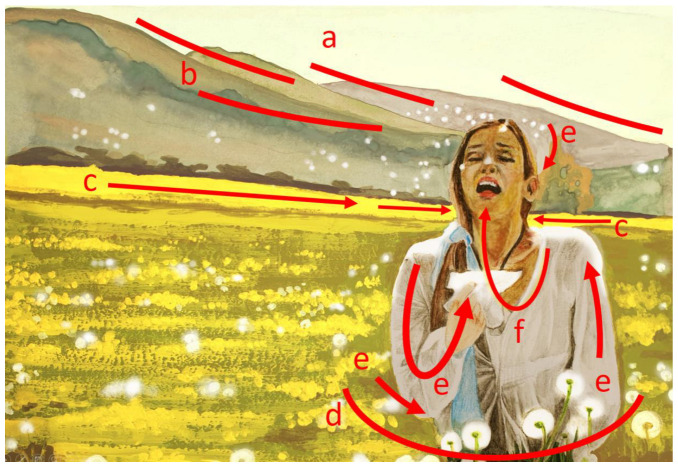
Formal analysis of the most frequently used picture in advertisements of allergy-related products and services depicting plants. The picture represents a popular design concept in visual marketing in allergies. Red arrows, lines and shapes are tools of formal analysis, showing how the viewer’s eyes are led by major elements and design principles (underlined in the text below) of the picture to the central figure (patient) and affected organs (mouth and nose). Note that the predominant color of the flowers in both images is yellow. The picture depicts a woman standing on a dandelion (*Taraxacum officinale*) field. The negative space (represented by herbaceous weed population) is large, homogenous, colorful and contrasting to the patient. The plants’ airborne cypselae fruits and sneezing suggests a causal relationship between the plant and the allergic reaction. The viewer’s eye is led by the lines of hills (a,b) as well as by colors, values and contrasts of the vegetation (c,d) to the central figure, whose movements (e,f) express a strong allergic reaction. The viewer’s impression (i.e., the effect of the artwork) is that the patient is ‘sinking into’ a field of allergenic weeds while sneezing and wheezing. The current picture (aquarelle painting) is made by the author (Donát Magyar) to represent the design of the original photograph [21].

**Figure 3 plants-12-01334-f003:**
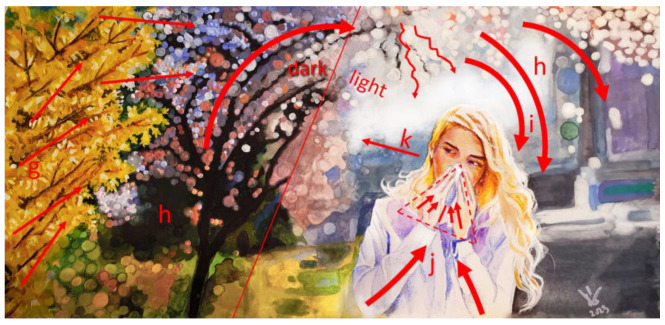
Formal analysis of a frequently used picture in advertisements of allergy-related products and services depicting plants. The picture represents a popular design concept in visual marketing in allergies. Red arrows, lines and shapes are tools of formal analysis, showing how the viewer’s eyes are led by major elements and design principles (underlined in the text below) of the picture to the central figure (patient) and affected organs (mouth and nose). The picture depicts a woman walking on a street, while the pollen of the trees (*Forsythia* × *intermedia* and Prunoidae) are ‘falling onto’ her, provoking an allergic reaction. The composition is unbalanced (weighted against the patient by the higher proportion of the dark area). Action is shown by lines of branches of ‘allergenic’ trees pointing to the patient (g,h), whose allergic reaction is emphasized by movements (j) and shapes (l, delineated by hands raised to the nose or mouth). A negative effect of plants is suggested by colors and values, as well as by lines, according to the direction of the patient’s gaze (k). The current picture (aquarelle painting) is made by the author (Donát Magyar) to represent the design of the original photograph [22].

**Figure 4 plants-12-01334-f004:**
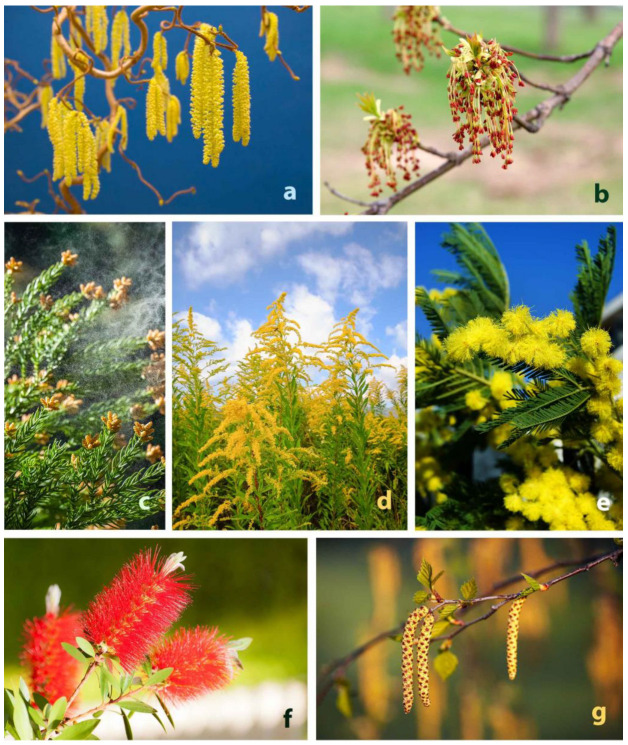
Recommended plant species for illustrating visual marketing materials of allergy-related products. (**a**): *Corylus avellana*, (**b**): *Acer negundo*, (**c**): *Cryptomeria japonica*, (**d**): *Solidago canadensis*, (**e**): *Acacia dealbata*, (**f**): *Callistemon* sp., (**g**): *Betula pendula*. Source: stock.adobe.com (accessed on 10 December 2022) (account NPHC).

**Table 1 plants-12-01334-t001:** Values of the CARE-S index and the corresponding levels of potential allergenicity. AC_p_ = raw calculated value of allergenicity categories [16].

AC_p_	Potential Allergenicity	CARE-S Index
0	non-allergenic	0
0 < AC_p_ ≤ 2	low	1
2 < AC_p_ ≤ 4	medium	2
4 < AC_p_ ≤ 6	high	3
6 < AC_p_ ≤ 8	very high	4
n.d.	no data; further studies needed	unknown

**Table 2 plants-12-01334-t002:** Plant taxa used in the illustrations of allergy-related marketing materials. Frequency: the % of the taxa; potential allergenicity according to the CARE-S index [16]: 0: non allergenic, 1: low, 2: medium, 3: high, 4: very high, n.d. = no data; W: woody, H: herbaceous, E: entomophilous flower; A: anemophilous flowers—catkin, B: other type of anemophilous flowers; Y: yellow, G: green, C: other bright-colored. * mostly *Crysanthemum* × *morifolium*, *C. maximum* and *C. coronarium*.

Taxa	Family	Frequency	Potential Allergenicity	Woody/Herbaceous	Pollination Type	Flower Color
*Acacia dealbata*	Fabaceae	0.4	2	W	0	Y
*Achillea filipendulina*	Asteraceae	0.2	1	H	0	Y
*Achillea millefolium*	Asteraceae	0.4	1	H	0	C
*Achillea ptarmica*	Asteraceae	0.2	1	H	0	C
*Alcea rosea*	Malvaceae	0.2	0	H	0	C
*Allium* sp.	Amaryllidaceae	0.5	0	H	0	C
*Alnus* spp.	Betulaceae	0.4	4	W	A	C
*Alstroemeria ligtu*	Alstroemeriaceae	1.2	1	H	0	C
*Ambrosia artemisiifolia*	Asteraceae	2.1	4	H	B	G
Anthemideae (cf. *Anthemis*)	Asteraceae	0.5	1	H	0	Y
Apiaceae	Apiaceae	2.0	0	H	0	C
*Aster* spp.	Asteraceae	0.4	2	H	0	C
Asteraceae	Asteraceae	3.4	1	H	0	C
*Begonia* hybrid	Begoniaceae	0.4	1	H	0	C
*Bellis perennis*	Asteraceae	1.1	1	H	0	C
*Betula* spp.	Betulaceae	0.9	4	W	A	Y
*Bougainvillea* spp.	Nyctaginaceae	1.1	0	H	0	C
*Brassica napus*	Brassicaceae	8.9	1	H	0	Y
Brassicaceae	Brassicaceae	0.4	1	H	0	Y
*Calendula officinalis*	Asteraceae	0.2	1	H	0	C
*Camellia japonica*	Theaceae	0.2	0	W	0	C
*Capsella bursa-pastoris*	Brassicaceae	0.2	0	H	0	C
*Carpinus* sp.	Betulaceae	0.2	2	W	0	C
*Ceiba speciosa*	Malvaceae	0.2	n.d.	W	0	C
*Centaurea* sp.	Asteraceae	0.5	1	H	0	C
*Cercis siliquastrum*	Fabaceae	0.2	0	W	0	C
*Chrysanthemum* spp.*	Asteraceae	6.4	1	H	0	C
Cichorieae	Asteraceae	0.4	1	H	0	C
*Cichorium intybus*	Asteraceae	0.4	1	H	0	C
*Cladanthus mixtus*	Asteraceae	0.7	n.d.	H	0	C
*Consolida regalis*	Ranunculaceae	0.2	0	H	0	C
*Coronilla varia*	Fabaceae	0.2	0	H	0	C
*Corylus avellana*	Betulaceae	0.7	4	W	A	Y
*Crocus* sp.	Iridaceae	0.2	n.d.	H	0	mostly Y
*Cytisus × praecox*	Fabaceae	0.5	1	W	0	Y
*Dactylis glomerata*	Poaceae	0.7	2	H	B	G
*Dahlia* sp.	Asteraceae	0.2	1	H	0	C
*Delphinium* sp.	Ranunculaceae	0.2	1	H	0	C
*Dianthus barbatus*	Caryophyllaceae	0.2	n.d.	H	0	C
*Dianthus caryophyllus*	Caryophyllaceae	0.2	1	H	0	C
*Dipsacus fullonum*	Caprifoliaceae	0.2	0	H	0	C
*Doronicum orientale*	Asteraceae	0.4	1	H	0	Y
*Epilobium angustifolium*	Onagraceae	0.2	0	H	0	C
*Erigeron annuus*	Asteraceae	0.2	n.d.	H	0	C
*Euphorbia pulcherrima*	Euphorbieae	0.2	0	W	0	C
*Eustoma exaltatum* ssp. *Russellianum*	Chironieae	0.2	1	H	0	C
*Eustoma grandiflorum*	Chironieae	0.7	1	H	0	C
*Forsythia × intermedia*	Oleaceae	2.3	n.d.	W	0	Y
*Genista* sp.	Fabaceae	0.2	0	H	0	C
*Gerbera hybrida*	Asteraceae	2.5	1	H	0	C
*Gladiolus communis* subsp. *Byzantinus*	Iridaceae	0.2	n.d.	H	0	C
*Gladiolus* hybrid (e.g., ‘Nova Lux’)	Iridaceae	0.2	1	H	0	C
*Gladiolus* × *gandavensis* ‘Bangladesh’	Iridaceae	0.4	1	H	0	C
*Grevillea alpina* ‘Olympic Flame’	Proteaceae	0.2	0	W	0	C
*Gypsophila paniculata*	Caryophyllaceae	0.2	n.d.	H	0	C
*Helianthus annuus*	Asteraceae	0.7	1	H	0	Y
*Helianthus* ‘Lemon Queen’	Asteraceae	0.5	1	H	0	Y
*Hibiscus syriacus*	Malvaceae	0.2	0	W	0	C
*Hydrangea arborescens*	Hydrangeaceae	0.2	0	H	0	C
*Iris* × *germanica*	Iridaceae	0.4	0	H	0	C
*Kalanchoë blossfeldiana*	Crassulaceae	0.7	0	H	0	Y
*Kerria japonica*	Rosaceae	0.5	0	W	0	Y
*Knautia* cf. *tatarica*	Caprifoliaceae	0.2	0	H	0	C
*Kolkwitzia amabilis*	Caprifoliaceae	0.2	0	W	0	C
*Laburnum anagyroides/L. × watereri*	Fabaceae	0.2	0	W	0	Y
*Lamium purpureum*	Lamiaceae	0.5	0	H	0	C
*Lavandula angustifolia*	Lamiaceae	0.7	0	H	0	C
*Leucanthemum maximum*	Asteraceae	0.2	1	H	0	Y
*Leucanthemum vulgare*	Asteraceae	2.7	1	H	0	C
*Lilium* sp. (hybrid)	Liliaceae	1.8	1	H	0	C
*Limonium sp.*	Plumbaginaceae	0.2	1	H	0	C
*Lupinus polyphillus*	Fabaceae	0.2	1	H	0	C
*Maloideae*	Rosaceae	1.2	0	W	0	C
*Mandevilla sanderi*	Apocynaceae	0.2	0	H	0	C
*Narcissus pseudonarcissus*	Amaryllidaceae	0.9	0	H	0	Y
*Nerium oleander*	Apocynaceae	0.5	0	W	0	C
*Papaver rhoeas*	Papaveraceae	0.4	0	H	0	C
*Pennisetum orientale*	Poaceae	0.2	2	H	B	G
*Pericallis × hybrida*	Asteraceae	0.2	1	H	0	C
*Petunia × hybrida*	Solanaceae	0.7	0	H	0	C
*Phalaenopsis hybrid*	Orchideaceae	0.4	0	H	0	C
*Philadelphus coronarius*	Hydrangeaceae	0.2	0	W	0	C
*Phlox diverticata*	Polemoniaceae	0.2	0	H	0	C
*Phlox paniculata*	Polemoniaceae	0.2	0	H	0	C
*Physostegia virginiana*	Lamiaceae	0.5	0	H	0	C
Poaceae	Poaceae	1.4	2	H	B	G
*Primula vulgaris*	Primulaceae	0.2	0	H	0	Y
Prunoideae	Rosaceae	13.0	0	W	0	C
*Prunus padus*	Rosaceae	0.2	0	W	0	C
*Prunus serrulata* ‘Kanzan’	Rosaceae	0.2	0	W	0	C
*Prunus triloba*	Rosaceae	0.2	0	W	0	C
*Ranunculus* sp.	Ranunculaceae	0.7	1	H	0	Y
*Rhododendron* sp.	Ericaceae	0.2	0	W	0	C
*Rosa* sp.	Rosaceae	3.6	1	W	0	C
*Rumex* sp.	Polygonaceae	0.2	2	H	B	G
*Salix* spp.	Salicaceae	0.9	n.d.	W	A	Y
*Salvia* cf. *pratensis*	Lamiaceae	0.4	0	H	0	C
*Salvia splendens*	Lamiaceae	0.2	0	H	0	C
*Scabiosa/Succisa/Knautia* sp.	Caprifoliaceae	0.2	0	H	0	C
*Senecio* spp.	Asteraceae	0.2	1	H	0	Y
*Silene flos-cuculi*	Caryophyllaceae	0.2	0	H	0	C
*Silene vulgaris*	Caryophyllaceae	0.2	0	H	0	C
*Solidago canadensis*	Asteraceae	0.9	2	H	0	Y
*Solidago virgaurea*	Asteraceae	0.5	2	H	0	Y
*Sorbus* cf. *aria*	Rosaceae	0.2	0	W	0	C
*Spartium junceum*	Fabaceae	0.2	0	W	0	C
*Spiraea* sp.	Rosaceae	0.2	0	W	0	C
*Syringa vulgaris*	Oleaceae	3.0	1	W	0	C
*Tamarix* sp.	Tamaricaceae	0.5	2	W	0	C
*Tanacetum vulgare*	Asteraceae	0.5	1	H	0	Y
*Taraxacum officinale*	Asteraceae	6.2	1	H	0	Y
*Trifolium pratense*	Fabaceae	0.7	1	H	0	C
*Trifolium repens*	Fabaceae	0.4	1	H	0	C
*Tulipa* spp.	Liliaceae	2.0	0	H	0	mostly Y
*Verbascum* sp.	Scrophulariaceae	0.2	0	H	0	Y
*Veronica longifolia*	Plantaginaceae	0.4	0	H	0	C
*Viburnum × burkwoodii*	Adoxaceae	0.4	1	W	0	C
*Viburnum opulus* ‘Roseum’	Adoxaceae	0.2	0	W	0	C
*Viburnum rhytidophyllum*	Adoxaceae	0.2	n.d.	W	0	C
*Vicia villosa*	Fabaceae	0.2	0	H	0	C
*Viola wittrockiana*	Violaceae	0.2	0	H	0	C
*Wisteria sinensis*	Fabaceae	0.4	0	W	0	C
*Xeranthemum annuum*	Asteraceae	0.2	0	H	0	C

**Table 3 plants-12-01334-t003:** Plant species recommended for illustrating visual marketing materials of allergy-related products.

Latin Name	English Name	Flower Color	Time of Interest (Pollination Period)	Area of Interest (Bioclimatic Region)	Potential Allergenicity (CARE-S Index)	Invasive
*Acacia dealbata*	silver wattle, blue wattle or mimosa	yellow	January-March	mostly warm regions; Australia (native), California, New Zealand, Portugal, South Africa, Spain	2	yes
*Acer negundo*	box elder, Manitoba maple	green, pink and red	April-May	mostly temperate regions; North America (native), South America, Australia, much of Europe, New Zealand, South Africa, parts of Asia	4	yes
*Betula* spp. (*B. ermanii*, *B. papyrifera*, *B. pendula*, *B. pubescens*)	birch	yellow	March-May	mostly temperate and cold regions	4	
*Callistemon*/*Melaleuca* spp.	bottlebrush	red	October-May	Western Australia (native), temperate regions	2	
*Corylus* spp. (*C. avellana*, *C. corulna*)	hazel	yellow	January-April	mostly temperate and cold regions	4	
*Cryptomeria japonica*	sugi, Japanese cedar, Japanese redwood	orange	February-March	mostly in Japan, China, Azores	4	
*Solidago canadensis*	Canadian goldenrod	yellow	July-October	mostly temperate and cold regions; North America (native), Europe, Japan and China	2	yes

## Data Availability

The data presented in this study are available on request from the corresponding author.

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
