# Peer review of "Potential Allergenicity of Plants Used in Allergological Communication: An Untapped Tool for Prevention"

_plants, 2023, doi:10.3390/plants12061334_

Round 1
Reviewer 1 Report
The article deals with a topical issue. The author of the article linked the research area with the marketing area. It has a considerable overlap into practice. The benefit is the use of the findings in practice (drug manufacturers, advertisement makers, customers, etc.) The author has done a lot of theoretical work. The methodology is described in a clear manner and the results are presented clearly. The discussion is handled at a good level.
Author Response
Thank you very much for the positive review.
Reviewer 2 Report
Read the manuscript and author tried to explore the allergenic plants advertisement usign online photographs. methods are well presented and conclusion supports the idea. However my question is is the photographs are enough to conclude such a study? even the author presented photographs are not enough to conclude are there any test author did to convince author him/herself?? Did the plant listed are actually have allergenic pollen or does the plant produce a mass pollen which can travel long distance with the flower of air current? This dynamics are not clear enough.
Author Response
Reviewer’s comment: Read the manuscript and author tried to explore the allergenic plants advertisement usign online photographs. methods are well presented and conclusion supports the idea. However my question is is the photographs are enough to conclude such a study? even the author presented photographs are not enough to conclude are there any test author did to convince author him/herself??
Answer: Thank you for your review. As the results of the relevant allergological tests are available in the medical literature, we reviewed and cited them in the Supplement Table 1, and, for a clearer interpretation we added the following text to the manuscript: “The allergological literature was reviewed for IM”
Reviewer’s comment: Did the plant listed are actually have allergenic pollen or does the plant produce a mass pollen which can travel long distance with the flower of air current? This dynamics are not clear enough.
Answer: Thank you for this question. The current study is based on our previous paper (Magyar, D., Páldy, A., Szigeti, T., & Orlóci, L. 2022. A regulation-oriented approach for allergenicity categorization of plants. Urban Forestry & Urban Greening, 70, 127530.), where the difference in actual and potential allergenicity is defined. In brief, potential allergenicity is an index made to express factors genetically determined in plants, which are independent of geographical area (e.g., immunogenicity, morphology, pollen production) versus geographically determined factors (e.g., length of the pollen season, population size of the plant, pollen concentration, etc.). Pollen mass production and long distance transport are factors depending on to the flower and pollen morphology, thus potential allergenicity. For example, birch pollen can be produced in high quantity in anemophilous flowers and transported by wind currents to long distances. On the other hand, Rosa spp. Produces relatively fewer pollen grains in the entomophilous flowers, which are not windborne. Potential allergenicity is therefore determined genetically even if the plant is introduced to a new area. For a better understanding on why the separation of potential and actual allergenicity is necessary, the following example should be considered. Olea europea has a high abundance and actual (epidemiological) allergenicity in the Mediterranean regions, but not in the Pannonian Region, where this plant is not present. Recently, this tree has been sold as a potted plant for decoration purposes. Based on its potential allergenicity, trade of such plants should be suppressed. If an allergenic categorization system were used based only on epidemiological allergenicity new, potentially allergenic taxa would be allowed for plantation, because local plant abundance is nil or still low, and allergy is not present in the local human population. However, a system based on potential allergenicity is able to suppress the commercial spread of this allergenic species in its new habitat. For a better readability, the following explanation were added to the manuscript: “Morphology of male flowers and pollen is considered to categorize plants according to their pollination strategy. Light, small (usually >40µm) and dry pollen grains with low specific gravity, or with air sacs having (nearly) smooth surface were considered to be anemophilous. Pollen production of culti-vars having high or reduced pollen yield were also considered in the categorization.”
Reviewer 3 Report
The article “Potential Allergenicity of Plants Used in Allergological Communication: an Untapped Tool for Prevention” is focused on Allergological Communication of the region that is very uncommon research. From this point of view, the subject of the study is very interesting and significant. The research topic is of some interest due to the current concern with the plant pollen on allergic reactions. The obtained results from this study will be better able to provide more useful information for understanding the release of plant pollen on allergic reactions. Nevertheless, minor revisions are required at the reviewer opinion.
1. Because of lacking detailed vegetation surveys, the authors should give details information of the vegetation in the study area. They also should provide information about the larger vegetation zone. If possible please add one figure which can show the information about vegetation coverage.
2. The objectives should write clearly, it seems a little confusing. Please rewrite it.
3. Research becomes very weak if you use only the internet data. Next time try to gather your own data for research.
4. It is essential to write all the pollen taxa in the result section which were identified in this study period. Moreover, no information about which pollen is more allergenic. What type of benefit allergenic people will found from the paper?
5. Conclusions should be more related to the result. Please focus on some benefits for allergenic people.
Author Response
The article “Potential Allergenicity of Plants Used in Allergological Communication: an Untapped Tool for Prevention” is focused on Allergological Communication of the region that is very uncommon research. From this point of view, the subject of the study is very interesting and significant. The research topic is of some interest due to the current concern with the plant pollen on allergic reactions. The obtained results from this study will be better able to provide more useful information for understanding the release of plant pollen on allergic reactions. Nevertheless, minor revisions are required at the reviewer opinion.
Reviewer’s comment: Because of lacking detailed vegetation surveys, the authors should give details information of the vegetation in the study area. They also should provide information about the larger vegetation zone. If possible please add one figure which can show the information about vegetation coverage.
Answer: Thank you for for your review. As the pictures were collected from the Internet, there are no direct indication of the original vegetation zone. Some of the presented ornamental plants probably were produced in greenhouses, thus they were propagated independently of outdoor climatic factors. However, it worth to mention that the current study included plants of various biogeographical regions.
Reviewer’s comment: The objectives should write clearly, it seems a little confusing. Please rewrite it.
Answer: Thank you for this suggestions. The objectives were rewritten. The new text is :” The aim of this study is to evaluate whether the images provided by allergy-related websites are suitable to convey appropriate information about the allergenicity of plants.“
Reviewer’s comment: Research becomes very weak if you use only the internet data. Next time try to gather your own data for research.
Answer: Thank you for this suggestion. Non-internet based materials, e.g. printed leaflets, journals and packaging materials depicting plants, as well as videos and ads in TV can be also collected as data source in the future.
Reviewer’s comment: It is essential to write all the pollen taxa in the result section which were identified in this study period.
Answer: Thank you for this correction. All the pollen taxa of this study is mentioned in Supplement table 1, column F.
Reviewer’s comment: Moreover, no information about which pollen is more allergenic.
Answer: Thank you for this remark. Our potential allergenicity index (CARE-S) contains information about immunogenicity (Supplement table 1, column B). It is important to emphasize that our approach in CARE-S is to rank immunogenicity by the strength of evidence, not by the strength of symptoms. Apparently, it would be difficult to scale the strength of immunogenicity of plant taxa, because this information comes from different diagnostic methods (allergen-specific IgE, Prick, roentgen diffraction, etc.). A literature review was conducted about the immunogenicity of each plant taxon. The research was performed in the article collections of the Allergome database (http://www.allergome.org) and other sources.
Reviewer’s comment: What type of benefit allergenic people will found from the paper?
Answer: Hopefully, this study will help to push back false information presented in allergy-related advertisements, and correct pictures will be used on them. This will help in patient education to avoid allergenic plants, and reduce exposure to allergenic pollen.
Reviewer’s comment: Conclusions should be more related to the result. Please focus on some benefits for allergenic people.
Answer: Thank you for this suggestion. The following text was added to the conclusions: “Correct illustrations will help patients to recognize and avoid allergenic plants, and reduce exposure to allergenic pollen”